# SARS-CoV-2 seroprevalence and neutralizing activity in donor and patient blood

Dianna L. Ng[1,2,13], Gregory M. Goldgof[1,13], Brian R. Shy[1,13], Andrew G. Levine[1,13], Joanna Balcerek[1,13], Sagar P. Bapat[1,13], John Prostko[3], Mary Rodgers[3], Kelly Coller[3], Sandra Pearce[3], Sergej Franz[4], Li Du[4], Mars Stone[1,4], Satish K. Pillai[4], Alicia Sotomayor-Gonzalez[1,5], Venice Servellita[1,5], Claudia Sanchez San Martin[1,5], Andrea Granados[1,5], Dustin R. Glasner [1,5], Lucy M. Han[1,2], Kent Truong[1,2], Naomi Akagi[1,2], David N. Nguyen [6], Neil M. Neumann [2], Daniel Qazi[2], Elaine Hsu[1], Wei Gu[1], Yale A. Santos[1,5], Brian Custer[4], Valerie Green[7], Phillip Williamson[7], Nancy K. Hills[8,9], Chuanyi M. Lu[1,10], Jeffrey D. Whitman [1], Susan L. Stramer[11], Candace Wang[1,5], Kevin Reyes[1,5], Jill M. C. Hakim[12], Kirk Sujishi[1], Fariba Alazzeh[1], Lori Pham[1], Edward Thornborrow[1], Ching-Ying Oon[1], Steve Miller[1,5], Theodore Kurtz [1], Graham Simmons[1,4,14], John Hackett Jr.[3,14], Michael P. Busch [1,4,14] & Charles Y. Chiu [1,5,6,14✉]

Given the limited availability of serological testing to date, the seroprevalence of SARS-CoV-2-specific antibodies in different populations has remained unclear. Here, we report very low SARS-CoV-2 seroprevalence in two San Francisco Bay Area populations. Seroreactivity was 0.26% in 387 hospitalized patients admitted for non-respiratory indications and 0.1% in 1,000 blood donors in early April 2020. We additionally describe the longitudinal dynamics of immunoglobulin-G (IgG), immunoglobulin-M (IgM), and in vitro neutralizing antibody titers in COVID-19 patients. The median time to seroconversion ranged from 10.3–11.0 days for these 3 assays. Neutralizing antibodies rose in tandem with immunoglobulin titers following symptom onset, and positive percent agreement between detection of IgG and neutralizing titers was >93%. These findings emphasize the importance of using highly accurate tests for surveillance studies in low-prevalence populations, and provide evidence that seroreactivity using SARS-CoV-2 anti-nucleocapsid protein IgG and anti-spike IgM assays are generally predictive of in vitro neutralizing capacity.

[1] Department of Laboratory Medicine, University of California, San Francisco, San Francisco, CA, USA. [2] Department of Pathology, University of California, San Francisco, San Francisco, CA, USA. [3] Applied Research and Technology, Abbott Diagnostics, Abbott Park, IL, USA. [4] Vitalant Research Institute, San Francisco, CA, USA. [5] UCSF-Abbott Viral Diagnostics and Discovery Center, San Francisco, CA, USA. [6] Department of Medicine, Division of Infectious Diseases, University of California, San Francisco, San Francisco, CA, USA. [7] Creative Testing Solutions, Tempe, AZ, USA. [8] Department of Neurology, University of California, San Francisco, San Francisco, CA, USA. [9] Department of Epidemiology and Biostatistics, University of California, San Francisco, San Francisco, CA, USA. [10] Laboratory Medicine Service, San Francisco VA Health Care System, San Francisco, CA, USA. [11] American Red Cross, Gaithersburg, MD, USA. [12] Department of Medicine at ZSFG, The Division of HIV, ID & Global Medicine, San Francisco, CA, USA. [13] These authors contributed equally: Dianna L. Ng, Gregory M. Goldgof, Brian R. Shy, Andrew G. Levine, Joanna Balcerek, Sagar P. Bapat. [14] These authors jointly supervised: Graham Simmons, John Hackett, Jr., Michael P. Busch, Charles Y. Chiu. ✉email: charles.chiu@ucsf.edu

Coronavirus disease 2019 (COVID-19) is a novel respiratory illness caused by the severe acute respiratory syndrome coronavirus 2 (SARS-CoV-2)[1]. The symptoms of COVID-19 range from asymptomatic infection to acute respiratory distress syndrome and death, and the COVID-19 pandemic has resulted in substantial burdens on healthcare systems worldwide[2,3]. Accurate and large-scale serologic testing that includes detection of neutralizing antibodies is essential in evaluating spread of infection in the community, informing public health containment efforts, and identifying donors for convalescent plasma therapy trials. Given the current state of diagnostic testing which largely relies on molecular techniques, the seroprevalence of SARS-CoV-2-specific antibodies—a proxy for prior infection—in different populations has remained unclear.

Here, we present data validating the use of the EUA authorized Abbott Architect SARS-CoV-2 IgG test for antibody detection in two populations in March 2020, a hospitalized COVID-19 patient cohort at a tertiary care hospital in San Francisco and a cohort of blood donors from the San Francisco Bay Area. We also investigate the longitudinal dynamics of IgG, IgM, and in vitro neutralizing antibody titers in hospitalized COVID-19 patients over time. These studies demonstrate that SARS-CoV-2 seroprevalence in the San Francisco Bay Area was very low, suggesting limited circulation of the virus in the community as of early March, and that IgG and IgM titers are predictive of neutralizing activity, with high positive percent agreement.

## Results

### Performance characteristics of the Abbott Architect SARS-CoV-2 IgG and IgM assays

Prior to assessing seroprevalence of SARS-CoV-2 antibodies in San Francisco Bay area populations, we verified the performance of the Abbott Architect SARS-CoV-2 IgG (FDA Emergency Use Authorization (EUA)) and IgM (prototype) assays. These assays are chemiluminescent microparticle immunoassays that target the nucleocapsid and spike proteins, respectively. The nucleocapsid protein was targeted in the Architect IgG assay as it was found to have increased sensitivity compared to the spike protein[4–6]. To evaluate assay sensitivity, we assembled a cohort of 38 hospitalized patients and 5 outpatients at University of California, San Francisco (UCSF) Medical Center and the San Francisco Veterans Affairs (SFVA) Health Care System, all of whom received care at adult inpatient units or clinics and were real-time polymerase chain reaction (RT-PCR) positive for SARS-CoV-2 from nasopharyngeal and/or oropharyngeal swab testing (Fig. 1a and Supplementary Table 1). The percentage of patients seroconverting for IgG at weekly time intervals following reported symptom onset reached 94.4% at ≥22 days (Fig. 1b, left). Correspondingly, IgG assay sensitivity from analysis of all 423 samples increased weekly to reach 96.9% at ≥22 days, and was 99% when samples from seven immuno-compromised patients (see below) were excluded (Fig. 1b, right, and Table 1). The percentage of patients seroconverting for IgM was also 94.4% at ≥22 days (Fig. 1c, left) and IgM assay sensitivity from analysis of 346 samples was 97.9% (98.9% with immunocompromised patients excluded) (Fig. 1c, right, and Table 1). To evaluate assay specificity, serum and plasma samples collected by Abbott Laboratories from US blood donors from Miami, Florida prior to the COVID-19 pandemic ("pre-COVID-19") were tested for IgG ($n = 1013$) and IgM ($n = 1492$) antibody seroreactivity. Two samples out of 1013 were positive by IgG testing, yielding an IgG specificity of 99.8% (95% CI: 99.3–100%) (Fig. 1d, top), concordant with the 99.9% specificity reported in an independent study by the University of Washington[7,8]. Six samples out of 1492 from US blood donors were positive by IgM testing, yielding a IgM specificity of 99.6% (95% CI: 99.2–99.9%)

(Fig. 1d, bottom). Thus, the Architect SARS-CoV-2 IgG and IgM assays demonstrated high sensitivity (96.9% at ≥22 days in a primarily hospitalized patient cohort) and specificity (99.6–99.8% % in pre-COVID blood donors).

### Longitudinal dynamics of IgG and IgM titers in COVID-19 patients

Given the availability of longitudinal samples from our cohort of 43 SARS-CoV-2 PCR positive patients, we next analyzed the longitudinal dynamics of plasma IgG (286 samples) and IgM (249 samples) titers during the course of hospitalization (Fig. 1e, f). Anti-nucleocapsid IgG and anti-spike IgM antibody titers were observed to rise approximately in tandem, with similar median times to seroconversion of 11.0 and 10.8 days, respectively. These results are consistent with a previous study reporting concomitant detection of IgG and IgM antibodies to spike protein, or, in some instances, earlier detection of IgG antibodies than IgM in COVID-19 patients[9–11]. Thus, it is possible that sequential IgM followed by IgG production may not be a general feature of the immune response to SARS-CoV-2, as is the case for many viral infections. In addition, the differences in time to seroconversion may be related to biologic variability among patients. It is also possible that some of observed variability and early seroconversion may be a result of initially mild disease symptoms leading patients to self-report delayed symptom onset dates.

Of the four patients in week 3 who had not yet seroconverted for IgG (Fig. 1b, left, days 15–21), two were kidney transplant recipients on tacrolimus and mycophenolate mofetil (MMF) immunosuppressive therapy; one was >90 years old; and one was an asymptomatic patient receiving acute psychiatric care who provided an unreliable history. Both renal transplant recipients were observed to ultimately seroconvert for IgG and IgM. Notably, delayed seroconversion for IgG and IgM was not universal among immunosuppressed patients. Three additional solid organ transplant (SOT) recipients on tacrolimus and MMF, as well as one patient with rheumatoid arthritis on methotrexate and infliximab, all seroconverted within 2 weeks (Fig. 1e, f), while another SOT recipient was positive for IgG and IgM in the earliest available serum sample from day 17 post symptom onset. We did not have samples beyond day 18 for the remaining two patients who were not immunosuppressed; however, as seroconversion was observed as late as three weeks after symptom onset (Fig. 1e, f), it is possible that analysis of later samples would have demonstrated detectable antibodies in their serum. The one patient who was still IgG negative in the 22+ day time frame (Fig. 1b, left) from a day 29 plasma sample had only mild symptoms and was positive by IgM and neutralizing antibody testing (described below). Conversely, the only IgM negative case in the 22+ day time frame (Fig. 1c, left) was IgM negative from a day 50 plasma sample but IgG positive, likely due to waning of IgM antibody titers to undetectable levels by this later time point.

To further evaluate assay specificity, we tested 235 remnant plasma samples from 163 SARS-CoV-2 PCR-negative UCSF patients collected from late March to early April 2020. The testing resulted in detection of only one reactive sample, yielding a specificity of 99.6% (95% CI: 97.7–100%) (Fig. 2a, 2nd column). The IgG reactive sample was from a patient admitted for syncope but who reported a cough of 1-month duration, suggesting potential prior infection from either SARS-CoV-2 or another human coronavirus that may have elicited a cross-reactive antibody response. We also tested 39 SARS-CoV-2 PCR negative UCSF patients for IgM antibody, none of whom were positive (Fig. 2b, 2nd column). These results of 99.6–100% specificity from SARS-CoV-2 PCR-negative UCSF patients are thus comparable to the 99.6–99.8% specificity of the Architect

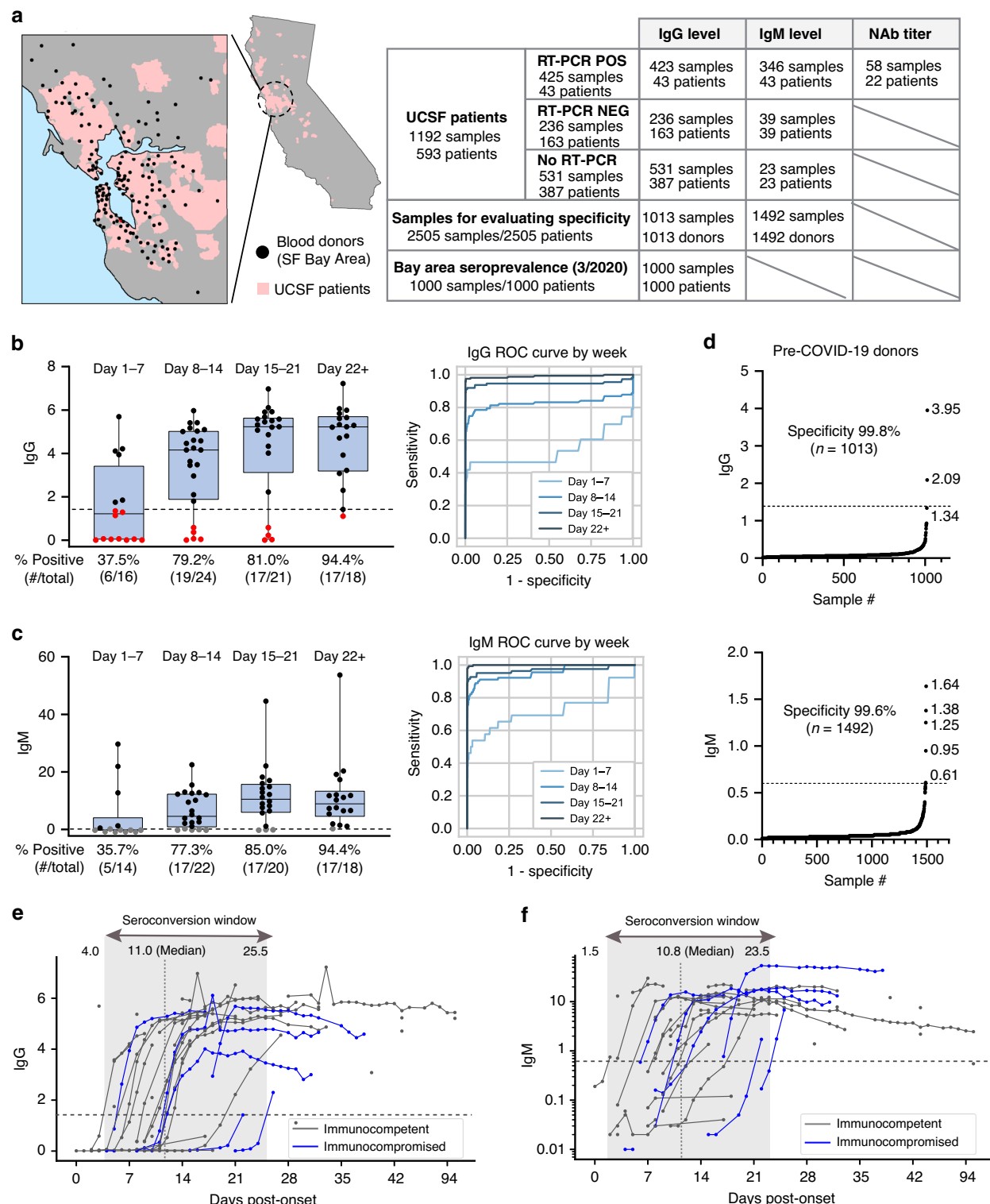

CoV-2 IgG and IgM assays calculated from pre-COVID-19 samples (Fig. 1d).

**Seroprevalence of SARS-CoV-2 in blood donors and patients from the San Francisco Bay Area.** To investigate SARS-CoV-2 seroprevalence in the San Francisco Bay Area, we performed anti-nucleocapsid IgG testing on plasma and serum samples from two cohorts of individuals with low suspicion of infection from COVID-19. One cohort consisted of 1000 individuals who donated blood in March 2020 at blood bank centers throughout the Bay Area (Fig. 1a and Supplementary Table 2). Routine blood donor screening was performed to exclude those with self-reported symptoms of acute illness and abnormal vital signs. We detected four IgG positive samples in this cohort, yielding a seroreactivity rate of 0.40% (Fig. 2a, 4th column). This cohort was not tested for IgM antibody. We then analyzed the four IgG positive samples using two orthogonal tests, the VITROS anti-SARS-CoV-2 total antibody assay (Ortho Clinical Diagnostics

**Fig. 1 Seroprevalence of antibodies to SARS-CoV-2. a** Schematic of testing performed and location of populations assessed. (Left) A map of California with a zoomed inset of the San Francisco Bay Area. For blood donors, the location of the blood bank collection center is denoted by a black dot; for patients at University of California, San Francisco (UCSF) hospitals and clinics, the geographic regions corresponding approximately to zip code are denoted in pink. (Right) Table showing number of patients, donors, and samples, and the testing that was performed (**b**) (left) IgG S/C ratios for SARS-CoV-2 PCR-positive patient samples for the indicated weekly timeframes post-onset of symptoms (if multiple samples per patient were collected, the sample with the highest S/C value within each time frame is plotted). The percent of patients with positive antibody responses measured within each timeframe is indicated below the graphs. (right) Receiver operating characteristic (ROC) curves for IgG titersx for all samples from SARS-CoV-2 PCR-positive patients within the indicated weekly time frames. AUCs for are 0.537 (day 1–7), 0.827 (day 8–14), 0.946 (day 15–21), 0.990 (day 22+). The dotted line at 1.4 indicates cutoff for IgG positivity. **c** IgM S/C ratios and ROC curves for IgM titers, as in **b**; AUCs are 0.720 (day 0–7), 0.955 (day 8–14), 0.970 (day 15–21), 0.999 (day 22+). The dotted line at 0.6 indicates cutoff for IgG positivity. Data points in black and red are above and below the indicated cutoffs, respectively. **d** (Top) IgG S/C ratios measured in pre-COVID samples; specificity and number of samples is indicated on graph (+). The dotted line at 1.4 indicates cutoff for IgG positivity. **d** (Bottom) IgM S/C ratios measured in pre-COVID samples; specificity and number of samples is indicated on graph. The dotted line at 0.6 indicates cutoff for IgM positivity. **e** IgG S/C ratios ($n = 286$ samples from 43 patients) and **f** IgM S/C ratios for SARS-CoV-2 PCR-positive patients ($n = 249$ samples from 43 patients) were plotted against day post symptom onset. Immunocompetent patients are shown in grey and immunocompromised patients are shown in blue. For patients with multiple same-day samples, the sample with the highest S/C value is plotted. For **e** and **f**, the box outlines denote the IQR, the solid line in the box denotes median S/C ratio, and the whiskers outside of the box extend to the minimum and maximum S/C ratios.

**Table 1 Clinical sensitivities of the Abbott Architect SARS-CoV-2 IgG and IgM and in vitro neutralization assays.**

Percentage of positive specimens from patients with positive SARS-CoV2 RT-PCR grouped by days since symptom onset and immune status

| Assay | All patient samples | | | | Immunocompetent only | | | | Immunocompromised only | | | |
|---|---|---|---|---|---|---|---|---|---|---|---|---|
| | Total $n$ | positive | % | 95% CI | Total $n$ | positive | % | 95% CI | Total $n$ | positive | % | 95% CI |
| Architect SARS-CoV-2 IgG | | | | | | | | | | | | |
| Day 1–7 | 41 | 12 | 29.3 | 23.7–35.6 | 35 | 10 | 28.6 | 22.5–35.5 | 6 | 2 | 33.3 | 16.1–55.3 |
| Day 8–14 | 106 | 68 | 64.2 | 60.5–67.7 | 82 | 53 | 64.6 | 60.4–68.7 | 24 | 15 | 62.5 | 53.5–70.7 |
| Day 15–21 | 113 | 102 | 90.3 | 87.7–92.3 | 77 | 72 | 93.5 | 90.5–95.6 | 36 | 30 | 83.3 | 77.1–88.1 |
| Day 22+ | 163 | 158 | 96.9 | 95.5–97.9 | 102 | 101 | 99 | 97.4–99.7 | 61 | 57 | 93.4 | 89.9–95.8 |
| All | 423 | 340 | 80.4 | 78.9–81.7 | 296 | 236 | 79.7 | 77.9–81.4 | 127 | 104 | 81.9 | 79.0–84.4 |
| Architect SARS-CoV-2 IgM | | | | | | | | | | | | |
| Day 1–7 | 26 | 10 | 38.5 | 30.6–47.0 | 22 | 9 | 40.9 | 32.1–50.4 | 4 | 1 | 25.0 | 6.9–54.4 |
| Day 8–14 | 91 | 68 | 74.7 | 70.9–78.1 | 70 | 54 | 77.1 | 72.8–80.9 | 21 | 14 | 66.7 | 56.9–75.2 |
| Day 15–21 | 83 | 75 | 90.4 | 87.2–92.8 | 53 | 49 | 92.5 | 88.4–95.2 | 30 | 26 | 86.7 | 79.9–91.5 |
| Day 22+ | 146 | 143 | 97.9 | 96.5–98.8 | 91 | 90 | 98.9 | 97.1–99.7 | 55 | 53 | 96.4 | 93.0–98.3 |
| All | 346 | 296 | 85.5 | 84.1–86.9 | 236 | 202 | 85.6 | 83.7–87.2 | 110 | 94 | 85.5 | 82.5–87.9 |
| Antibody neutralization assay | | | | | | | | | | | | |
| Day 1–7 | 10 | 4 | 40.0 | 26.1–55.5 | 9 | 3 | 33.3 | 19.6–50.2 | 1 | 1 | 100 | 25.0–100 |
| Day 8–14 | 24 | 14 | 58.3 | 49.4–66.8 | 18 | 12 | 66.7 | 55.9–76.0 | 6 | 2 | 33.3 | 16.1–55.3 |
| Day 15–21 | 10 | 7 | 70.0 | 54.2–82.4 | 6 | 5 | 83.3 | 61.1–95.3 | 4 | 2 | 50.0 | 24.3–75.7 |
| Day 22+ | 14 | 13 | 92.9 | 81.9–98.0 | 9 | 9 | 100 | 85.7–100 | 5 | 4 | 80.0 | 54.6–94.4 |
| All | 58 | 38 | 65.5 | 60.3–70.4 | 42 | 29 | 69.0 | 62.8–74.7 | 16 | 9 | 56.2 | 44.8–67.1 |

Clinical sensitivity of each assay, defined as the percent of samples from RT-PCR confirmed SARS-CoV-2 infected patients that test positive in each assay. Total numbers of samples, positive samples, and percent positive among total samples with 95% confidence intervals (CI) are shown for the indicated time frames for samples from all patients (left column), samples from immunocompetent patients only (middle column), and samples from immunocompromised patients only (right column). Immunocompromised patients: six solid organ transplant recipients on tacrolimus and MMF and one rheumatoid arthritis patient on methotrexate and infliximab.

EUA) and a SARS-CoV-2 pseudovirus neutralization assay (described below). Of the four samples, three (circled in Fig. 2a) were negative by both the VITROS and neutralization assays, and thus were designated likely false reactives by the Architect IgG assay. Thus, the calculated seroprevalence after confirmatory orthogonal testing for Bay Area blood donors in March 2020 was 0.1% (95% CI: 0.00–0.56%). The false reactive rate in this population of 0.3% is consistent with the reported specificity of the Architect SARS-CoV- IgG test of 99.6% (Fig. 1d)[7].

We additionally evaluated seroprevalence in a cross-section of patients who received care at adult inpatient units or clinics at the UCSF Medical Center for indications other than COVID-19 respiratory disease (non-COVID-19, never tested for SARS-CoV-2 by RT-PCR) from late March to early April 2020 (Supplementary Table 3). Remnant samples from 387 patients were obtained from UCSF clinical laboratories (Fig. 2a, 3rd column). Only one patient had IgG seroreactivity; this patient had presented with respiratory failure and ground-glass opacities on chest imaging but was never tested for SARS-2-CoV by RT-PCR. IgG seroprevalence in this population was thus low at 0.26% (95% CI: 0–0.76%), comparable to the 0.1%

seroprevalence in the 1000 tested Bay Area blood donors. Although only 23 of the remnant samples were able to be subsequently tested for IgM antibodies, importantly and as expected, none were positive (Fig. 2b, column 3).

Next, we sought to directly compare IgG and IgM antibody titers with SARS-CoV-2 in vitro neutralizing activity in 54 available plasma samples from 22 of the 43 SARS-CoV-2 PCR positive patients for whom residual longitudinal samples were available (Fig. 2c). Neutralizing antibody activity measured using SARS-CoV-2 pseudoviruses has previously been shown to correlate well with that measured using cultured SARS-CoV-2 isolates[12]. Plasma titers that achieved 80% neutralization of infectivity (NT80) using a SARS-CoV-2 pseudovirus, a vesicular stomatitis virus (VSV) pseudotype expressing the SARS-CoV-2 spike protein, were measured by luciferase assay (see "Methods"). The positive percent agreement (PPA) and negative percent agreement (NPA) were 92.5% and 93.7%, respectively, between IgG and IgM positivity; 93.8% and 75.0%, respectively, between NT80 and IgG positivity; and 84.8% and 78.6%, respectively, between NT80 and IgM positivity (Fig. 2c, left to right). All pairwise comparisons were linearly related with *rho* correlations

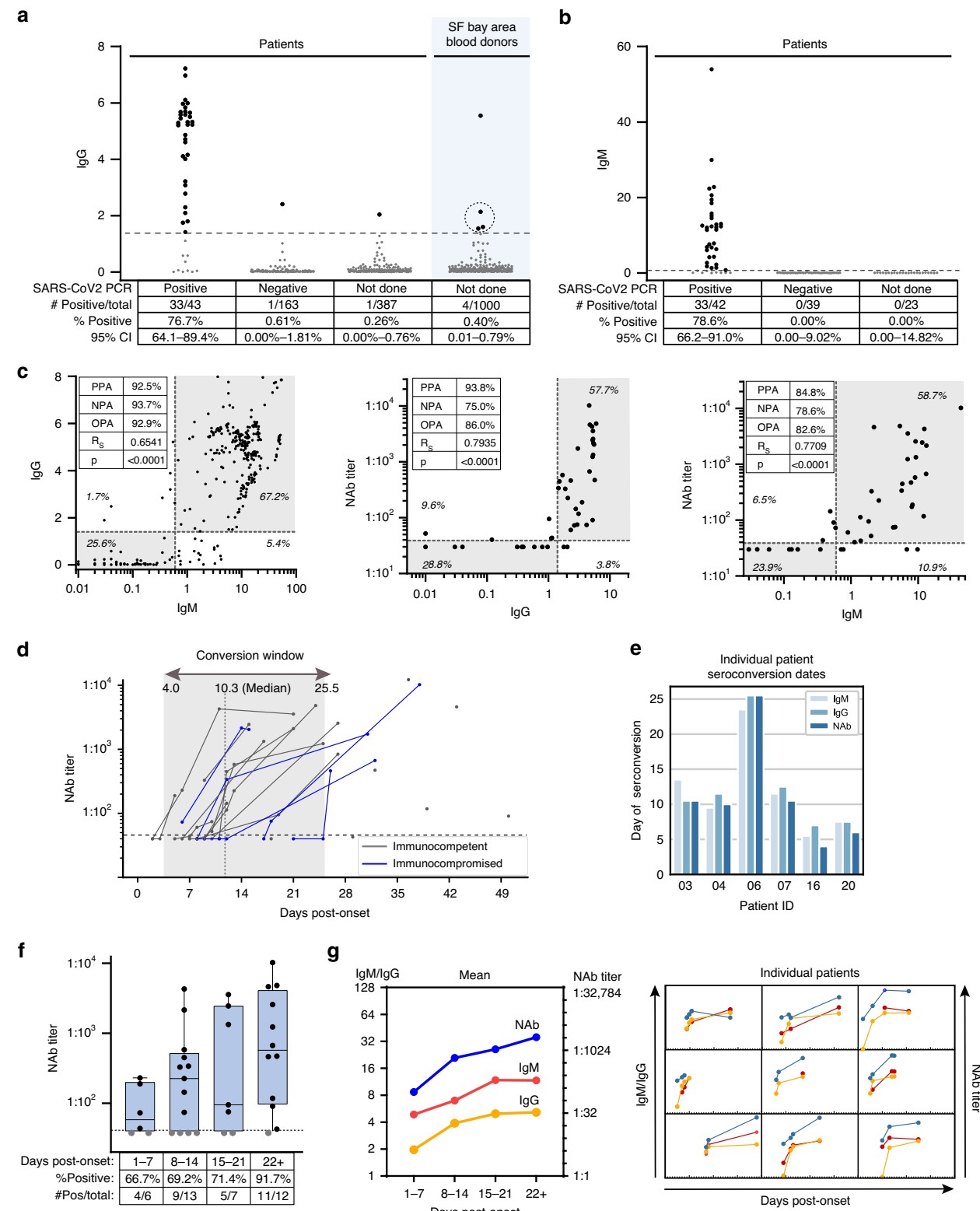

of 0.65, 0.79, and 0.77 between IgG and IgM positivity, NT80 and IgG positivity, and NT80 and IgM positivity, respectively. Neutralizing antibody titers rose in tandem with IgG and IgM antibody titers, with a slightly earlier median time of seroconversion of 10.3 days (versus 10.8–11.0 days for IgG/IgM antibody titers) (Fig. 2d–g). There was no significant correlation observed between median IgG, IgM, and neutralizing antibody titers and severity of disease (Supplementary Table 4).

## Discussion

In this study, we provide evidence that seropositive results using the Architect SARS-CoV-2 anti-nucleocapsid protein IgG and anti-spike IgM assays are generally predictive of in vitro neutralizing capacity. Given that anti-nucleocapsid antibodies are not thought to be directly neutralizing, these results suggest that detection of anti-nucleocapsid IgG antibodies may be indicative of a productive and polyclonal humoral immune response that

**Fig. 2 Longitudinal dynamics and in vitro neutralizing activity of antibodies against SARS-CoV-2. a** IgG S/C ratios were determined for hospitalized patients and outpatients and blood donors on whom SARS-CoV-2 PCR testing was positive or negative or was not performed. Numbers of seroreactive and total individuals tested are shown in tables below the graphs. The circled data points were additionally tested by the VITROS and neutralization assays and were negative by both assays. For patients with multiple samples, the single highest S/C value is plotted. **b** IgM S/C ratios, as in **a**. **c** (Left) IgG and IgM titers for SARS-CoV-2 PCR positive matched patient samples. Percent of data points in each quadrant and positive percent agreement (PPA), negative percent agreement (NPA), and overall percent agreement (OPA) between IgG and IgM are shown. (middle) 80% neutralization titers (NT80) plotted against IgG S/C values. (right) 80% NT80 plotted against IgM S/C values. The cutoff for NT80 is a titer level of >40; negative results (<40) are non-numeric and are plotted at 35 for visualization purposes. **d** NT80 titers for SARS-CoV-2 PCR-positive patients were plotted against day post symptom onset. Immunocompromised patients are shown in blue. **e** For the 6 SARS-CoV-2 PCR-positive patients whose IgM, IgG, and NT80 seroconversion events were captured during serial sampling, the days post-symptom onset seroconversion events are compared. **f** NT80 activity was evaluated per patient for the indicated time frames post onset of symptoms. The percent of patients with detectable NT80 activity measured within each time frame is indicated below the graphs. If multiple samples per patient were collected, the sample with the highest NT80 value within each time frame was used. **g** The average NT80 activity (right axis) and IgG and IgM (left axis) titers are plotted by day post-symptom onset (left); corresponding graphs for individual patients are shown in a 3 × 3 grid (right). If multiple samples per patient were collected, the sample with the highest S/C or NT80 value per time frame was used. For **f**, the box outlines denote the IQR, the solid line in the box denotes median neutralizing antibody titer, and the whiskers outside of the box extend to the minimum and maximum neutralizing antibody titers.

includes neutralizing (likely anti-spike) activity. Indeed, we also found high positive and NPA of >92% and good correlation ($rho = 0.65$) for detection of anti-nucleocapsid IgG and anti-spike IgM antibodies. These findings are also of relevance for serologic testing, as 44% (7 of 16) of the FDA EUA authorized serological assays as of June 2020 target the nucleocapsid protein, and also suggest that IgG and IgM titers may be predictive of neutralizing activity when identifying potential candidate donors for experimental convalescent plasma therapy. However, in vitro neutralization activity may not necessarily confer protective immunity and the efficacy of convalescent plasma therapy for treatment of COVID-19 disease remains to be determined.

Importantly, our results also show that the seroprevalence of IgG antibodies against SARS-CoV-2 in blood donors and non-COVID-19 patients seen at a tertiary care hospital in the San Francisco Bay Area from March to April 2020 is very low at 0.10% (95% CI: 0.00–0.56%) and 0.26% (0.00–0.76%), respectively. These seroprevalence rates in two distinct populations in the San Francisco Bay Area are near the specificity limit of the Architect assay, and are far lower than the specificity limits for many lateral flow immunoassays[13]. These findings contrast with those from other community-based studies that reported higher rates of seropositivity in California[14,15], and underscore the importance of using a highly accurate test for surveillance studies in low-prevalence populations. They also indicate a very low likelihood of widespread cryptic circulation of SARS-CoV-2 in the Bay Area prior to March 2020, consistent with the low detection rate by direct viral testing of respiratory samples collected during that early time period[16].

## Methods

**Study design and ethics**. The study population consisted of patients with available remnant serum and plasma specimens from the clinical laboratories at UCSF. Samples from patients who were positive or negative by SARS-CoV-2 RT-PCR testing of nasopharyngeal, oropharyngeal, and/or pooled nasopharyngeal–oropharyngeal swabs were collected in March–April 2020. Additional samples were collected from randomly selected cohorts of outpatients and hospitalized patients at UCSF during the same time period seen for indications other than COVID-19 respiratory disease (non-COVID). Serum samples from blood donors in the San Francisco Bay Area were collected by Vitalant Research Institute in March 2020. Clinical data for UCSF patients were extracted from electronic health records and entered in a HIPAA (Health Insurance Portability and Accountability Act)-secure REDCap research database. Collected data included demographics, major comorbidities, patient-reported symptom onset date, clinical symptoms and indicators of COVID-19 severity such as admission to the intensive care unit and requirement for mechanical ventilation. This study was approved by the institutional review board (IRB) at UCSF (UCSF IRB #10–02598) as a no-subject contact study with waiver of consent.

**Serologic testing**. The Abbott Architect SARS-CoV-2 IgG assay (FDA EUA) and SARS-CoV-2 IgM (prototype) testing was performed using either serum or plasma

samples on the Architect instrument according to the manufacturer instructions[7]. These tests are chemiluminescent microparticle immunoassay reactions that target the nucleocapsid protein (IgG assay) or the spike protein (IgM assay) and measure relative light units that are then used to calculate an index value. At a predefined index value threshold of 0.6 signal-to-cutoff (S/C) ratio for IgM seropositivity and 1.4 S/C for IgG for seropositivity, these assays were found to have specificities of 99.6–99.8%. The linear range is 1.4–4.0 S/C for the IgG assay, and 0.6–21 S/C for the IgM assay.

The VITROS anti-SARS-CoV-2 total antibody assay approved under FDA EUA was performed using either serum or plasma samples at Vitalant Research Institute according to the manufacturer instructions[15]. The test is a chemiluminescent immunoassay that targets the spike protein and measures relative light units that are then used to calculate an index value. At a predefined index value threshold of 1.0 signal-to-cutoff (S/C) ratio for IgG seropositivity, this assay was found to have a sensitivity of 100% (92.7–100%) and specificity of 100% (95% CI = 99.1–100.0%).

**Production of pseudoviruses for the SARS-CoV-2 neutralization assay**. VSVΔG-luciferase-based viruses, in which the glycoprotein (G) gene has been replaced with luciferase, were produced by transient transfection of viral glycoprotein expression plasmids (pCG SARS-CoV-2 Spike, provided courtesy of Stefan Pölhmann[17], as well as pCAGGS VSV-G or pCAGGS EboGP as controls[18]) or no glycoprotein controls into HEK293T cells by TransIT-2020. Briefly, cells were seeded into 15-cm culture dishes and allowed to attach for 24 h before transfection with 30 μg expression plasmid per plate. The transfection medium was changed at ~16 h post-transfection. The expression-enhancing reagent valproic acid was added to a final concentration of 3.75 mM, and the cells were incubated for 3–4 h. The medium was changed again, and the cells were inoculated with VSVΔG-luc virus at a multiplicity of infection of 0.3 for 4 h before the medium was changed again. At about 24 h post-infection, the supernatants were collected and cleared of debris by filtration through a 0.45-μm syringe filter.

**Antibody neutralization**. HEK293T cells were transfected with human ACE2 and TMPRSS2 by TransIT-2020[19]. After 24 h cells were plated into black 96-well tissue culture treated plates. Plasma, collected in lithium heparin tubes, was diluted to 1:20 followed by four subsequent 1:4 dilutions. Per well, 50 μl of pseudovirus harboring either SARS-CoV-2 S, VSV-G, or EboGP (adjusted to result in ~10,000 RLU in target cells) was mixed with 50 μl of the respective serum or plasma dilution to give a final series of longitudinal serum or plasma dilutions starting at 1:40 and incubated for 1 h at 37 °C. Controls included wells with VSVΔG (no envelope), without added serum/plasma, and with serum predetermined to possess or lack neutralizing activity. Subsequently, the 100 μl mix was added to the target cells (performed in duplicate) and cells were incubated for 24 h at 37 °C. Supernatants were then removed, cells were lysed, and luciferase activity was read as per manufacturer instructions. Results were calculated as a percentage of no serum control. Each plate was qualified by lack of infection with the no envelope control, and performance of positive and negative controls. Nonlinear regression curves and 80% neutralization titers (NT80) were calculated in GraphPad Prism v8.

**Statistical analysis**. We calculated PPA, NPA, and overall percent agreement (OPA) between the neutralizing antibody result and IgG, assuming IgG to be the gold standard. We then calculated PPA, NPA, and OPA between the neutralizing antibody result and IgM, assuming IgM to be the gold standard. We calculated 95% exact binomial (Clopper–Pearson) confidence intervals for each proportion. IgG, IgM, and NT80 titers were non-normally distributed and were summarized using medians and interquartile ranges. We compared antibody titers to dichotomously defined clinical characteristics at various time points using two-sided Wilcoxon

rank sum tests. The correlations between age and IgG, IgM, and NT titers were calculated using Spearman nonparametric correlation coefficients. Statistical calculations were performed using Python 3.7.7 using scipy.stats, sklearn.metrics.auc and statsmodels.stats libraries as well as Stata v15.1 (College Station, TX).

## Data availability

All data generated or analyzed during this study are included in this published article and associated Supplementary Information and Source Data files.

Source Data are provided with this paper.

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

## Acknowledgements

We thank the patients and their families at UCSF without whom collecting and providing this aggregate data would not have been possible. This work was funded by NIH grants R01-HL105704 (CYC) and R38HL143581 (G.M.G., S.P.B., and J.W.) from the National Heart, Lung, and Blood Institute, R33-AI129077 (CYC) the National Institute of Allergy and Infectious Diseases, the Charles and Helen Schwab Foundation (CYC). These funders had no role in study design, data collection and analysis, writing the paper, or decision to publish. This work was also funded in part by Abbott Laboratories. Employees of Abbott laboratories (J.P., M.R., K.C., S.P., J.H.) contributed to sample collection, IgG and IgM testing, and data analysis but had no role in the study design, writing the paper, or decision to publish.

## Author contributions

C.Y.C. conceived, designed, and supervised the study. D.L.N., and G.M.G. coordinated the study. D.L.N., G.M.G., B.R.S., A.G.L., S.P.B., J.B., and C.Y.C. analyzed data, designed figures, and wrote and edited the paper. A.S.G., V.S., C.S.S.M., A.G., D.R.G., E.H., W.G., Y.A.S., C.W., K.R., J.H., F.A., L.P., C.-Y.O., E.T., and C.M.L. contributed to the collection of clinical specimens. K.S., T.K., and E.T. coordinated clinical sample collection and IgG testing. S.M. provided clinical data and facilitated sample collection. L.M.H., K.T., N.A., D.N.N., N.M.N., and D.Q. performed chart review. M.S., B.C., V.G., P.W., M.B., and J.D.W. coordinated blood donor samples and data. S.F., L.D., G.S., and S.K.P. performed neutralizing antibody assays. J.M.C.H., J.P., M.R., K.C., and S.P. performed IgG and IgM testing and provided data establishing testing characteristics of SARS-CoV-2 IgG and IgM assays. N.K.H. performed biostatistical analysis and review. All authors read the paper and agreed to its contents.

## Competing interests

C.Y.C. is the director of the UCSF-Abbott Viral Diagnostics and Discovery Center (VDDC) and receives research support funding from Abbott Laboratories. J.P., M.R., K.C., S.P., and J.R.H., Jr. are employees of Abbott Laboratories. The other authors have no competing interests to declare.
