## [Peer Review File · Nature Communications]

REVIEWERS' COMMENTS:

Reviewer #1 (Remarks to the Author):

The authors have made substantial changes to the manuscript that overall address the reviewers' comments and concerns. The data in the figures have been rearranged to align with the presentation of the results in the text, which have improved the readability and flow.

Aspects that have been questioned have been satisfactorily clarified and additional details provided where requested. Of note, major queries pertaining to scientific rationale at various points throughout have been addressed and appropriately referenced where needed, including why nucleocapsid IgG antibodies were targeted, the choice of samples to assess neutralisation activity (based on availability of longitudinal samples) and substantiating the use of pseudoviruses to validate neutralisation.

While the authors have provided an explanation for "early (from d1.5/d4 after disease onset) seroconversion" citing similar observations made in other studies, this has not been specifically discussed in the text and should be included to place these observations into context.

Finally, with regards to the correlation of nucleocapsid-specific IgG antibodies with neutralisation, the authors have presented a valid explanation in that these antibodies are likely to parallel the rise of anti-spike IgM antibodies and highlight the correlation between these specificities and isotypes. If the authors can provide further correlation between nucleocapsid-specific IgG and spike-specific IgG responses, this would provide more evidence to support the claim that neutralisation is likely to be attributed to spike-specific responses.

RESPONSE TO REVIEWER COMMENTS

Please note that our responses are in boldfaced italics.

REVIEWERS' COMMENTS:

Reviewer #1 (Remarks to the Author):

The authors have made substantial changes to the manuscript that overall address the reviewers' comments and concerns. The data in the figures have been rearranged to align with the presentation of the results in the text, which have improved the readability and flow.

Aspects that have been questioned have been satisfactorily clarified and additional details provided where requested. Of note, major queries pertaining to scientific rationale at various points throughout have been addressed and appropriately referenced where needed, including why nucleocapsid IgG antibodies were targeted, the choice of samples to assess neutralisation activity (based on availability of longitudinal samples) and substantiating the use of pseudoviruses to validate neutralisation.

While the authors have provided an explanation for "early (from d1.5/d4 after disease onset) seroconversion" citing similar observations made in other studies, this has not been specifically discussed in the text and should be included to place these observations into context.

Thank you for the constrictive feedback, which has significantly improved this manuscript. Additional discussion of this important point has been included in the revised text:

"...In addition, the differences in time to seroconversion may be related to biologic variability among patients. It is also possible that some of observed variability and early seroconversion may be a result of initially mild disease symptoms leading patients to self-report delayed symptom onset dates..."

Finally, with regards to the correlation of nucleocapsid-specific IgG antibodies with neutralisation, the authors have presented a valid explanation in that these antibodies are likely to parallel the rise of anti-spike IgM antibodies and highlight the correlation between these specificities and isotypes. If the authors can provide further correlation between nucleocapsid-specific IgG and spike-specific IgG responses, this would provide more evidence to support the claim that neutralisation is likely to be attributed to spike-specific responses.

We agree that demonstrating the correlation between nucleocapsid-specific IgG and spike-specific IgG responses would provide more evidence. However, we regret that many of the samples from COVID-19 patients no longer have sufficient volume for additional serological testing. We note that several studies have described the correlation between nucleocapsid-specific IgG and spike-specific IgG responses, which are similar overall (Sun, B. et al. Emerg. Microbes Infect. 9, 940–948 (2020), Burbelo, P. D.

*et al. J. Infect. Dis. 222, 206–213 (2020), Liu, W. et al. J. Clin. Micro. 58, e00461-20 (2020).).
These studies are referenced in the revised manuscript.*